# Ochratoxin A-Induced Hepatotoxicity through Phase I and Phase II Reactions Regulated by AhR in Liver Cells

**DOI:** 10.3390/toxins11070377

**Published:** 2019-06-29

**Authors:** Hye Soo Shin, Hyun Jung Lee, Min Cheol Pyo, Dojin Ryu, Kwang-Won Lee

**Affiliations:** 1Department of Biotechnology, College of Life Sciences and Biotechnology, Korea University, Seoul 02841, Korea; 2School of Food Science, University of Idaho, 875 Perimeter Drive, Moscow, MS 2312, USA

**Keywords:** hepatotoxicity, ochratoxin A, oxidative stress, phase I reaction, phase II reaction

## Abstract

Ochratoxin A (OTA) is a widespread mycotoxin produced by several species of the genera *Aspergillus* and *Penicillium*. OTA exists in a variety of foods, including rice, oats, and coffee and is hepatotoxic, with a similar mode of action as aflatoxin B1. The precise mechanism of cytotoxicity is not yet known, but oxidative damage is suspected to contribute to its cytotoxic effects. In this study, human hepatocyte HepG2 cells were treated with various concentrations of OTA (5–500 nM) for 48 h. OTA triggered oxidative stress as demonstrated by glutathione depletion and increased reactive oxygen species, malondialdehyde level, and nitric oxide production. Apoptosis was observed with 500 nM OTA treatment. OTA increased both the mRNA and protein expression of phase I and II enzymes. The same results were observed in an in vivo study using ICR mice. Furthermore, the relationship between phase I and II enzymes was demonstrated by the knockdown of the aryl hydrocarbon receptor (AhR) and NF-E2-related factor 2 (Nrf2) with siRNA. Taken together, our results show that OTA induces oxidative stress through the phase I reaction regulated by AhR and induces apoptosis, and that the phase II reaction is activated by Nrf2 in the presence of oxidative stress.

## 1. Introduction

Ochratoxin A (OTA) is a mycotoxin produced by two genera of fungi: *Aspergillus* and *Penicillium* [1]. The most well-known species of *Aspergillus* producing OTA in food is *A. alutaceus*, which is predominantly present in green coffee beans, cocoa beans, beans, peanuts, rice and corn [2]. OTA is also produced at low frequency by *A. sulphureus*, *A. sclerotium* or *A. melleus* [3]. In genus *Penicillium*, only *P. verrucosum,* which is commonly associated with stored cereals and is frequently detected in northern Europe and Canada has been confirmed as OTA producer [4]. OTA is primarily found in oats, nuts, and coffee, which are consumed world-wide, and rice, which is a staple food for Asian people [5]. OTA is categorized as a carcinogen that is likely to be carcinogenic to humans, group 2B [6], and oxidative stress, DNA adducts, and signaling transduction were considered the causes of OTA toxicity [7]. In particular, OTA is known to have immunotoxicity, carcinogenicity, nephrotoxicity, hepatotoxic, neurotoxicity, and teratogenicity [2,8,9]. Because of these backgrounds, research on OTA is important. However, the toxicity of OTA in the liver has not been well studied compared with similar studies in the kidney, which is the main target organ of this drug [10]. 

The human hepatocyte cell line HepG2, which was used in this experiment, is a model cell line employed in toxicity studies [11] and is commonly used for drug metabolism and hepatotoxicity studies. The phase I reaction can lead to the oxidization of the absorbed xenobiotic compound into the original compound with a hydroxyl group. Oxidation reactions occur via the CYP family, and represent phase I reactions [12]. Among isoforms of the CYP450 enzymes, CYP1A1/1A2 and CYP3A4 expression is relatively high in the liver [13,14]. CYP1A1/1A2 is regulated by the aryl hydrocarbon receptor (AhR) [15], while CYP3A4 is regulated by the pregnane X receptor (PXR) [16]. When toxic substances enter the body, the transcriptional factors, AhR and PXR, migrate from the cytoplasm to the nucleus and increase the expression of CYP1A1/1A2 and CYP3A4, resulting in phase I oxidation [17]. The phase II reaction provides protection against oxidative stress, and antioxidant enzymes are involved in this process [18]. The main regulator of the antioxidant defense system is NF-E2-related factor 2 (Nrf2) [19], which controls the expression of various antioxidant enzymes, such as heme oxygenase-1 (HO-1) [20], glutamate-cysteine ligase (GCL) [21], and NAD(P)H quinone dehydrogenase 1 (NQO1) [22]. Under normal conditions, Nrf2 binds to Kelch-like ECH-associated protein 1 (keap-1), but, when activated by stress signals, it separates from keap1 and migrates to the nucleus [23] where it binds to the antioxidant response element (ARE) site of the target genes, including HO-1, glutamate-cysteine ligase catalytic subunit (GCLC), and NQO1, which counteract the effects of oxidative stress [23].

A landmark event of cellular self-destruction by apoptosis is the activation of nucleases, which lead to the degradation of nuclear DNA into fragments of approximately 200-base pairs long [24]. DNA breaks are labelled by terminal deoxynucleotide transferase dUTP nick end labeling (TUNEL). Since OTA induces oxidative stress resulting in DNA damage and apoptosis [25,26], the role of the phase I reaction in oxidative stress, and that of the phase II reaction, which is activated by the Nrf2-ARE signaling pathway, should be investigated in the presence of OTA. Therefore, we evaluated the expression of various oxidative stress-related markers, including reactive oxygen species (ROS), glutathione (GSH), malondialdehyde (MDA), nitric oxide (NO), and phase I/II enzymes after treatment with OTA in HepG2 cells. By silencing the expression of AhR and Nrf2, we were able to elucidate their relationship in phase I and II reactions.

## 2. Results

### 2.1. Effect of OTA on Viability, DNA Damage, and Apoptosis of HepG2 Cells

The 3-(4,5-Dimethyl-2-thiazolyl)-2,5-diphenyl-2H-tetrazolium bromide (MTT) assay was used to confirm the cytotoxicity of OTA (Figure 1a). Since OTA is chronic toxic, timecourse was performed to confirm cytotoxicity under low concentration conditions at long incubation times (24, 36 and 48 h). Cell viability decreased significantly with increasing OTA incubation time and concentration. Following treatment for 48 h, the IC_50_ value (IC_50_: concentration producing 50 % inhibition) (250 nM) was in the lower concentration range compared with that in other studies using OTA in the range of 5 to 1000 μM. Therefore, cells were treated with OTA for 48 h in all experiments. Following treatment with OTA at 5, 20, 100, 250, and 500 nM for 48 h, cell viability decreased in a dose-dependent manner. The cell viability was 50–60% of the untreated control at 250 nM OTA. Therefore, in this experiment, concentrations of 5 to 500 nM, which are lower and higher than the IC_50_, were chosen. 

OTA has been reported to induce DNA damage leading to apoptosis [27]. The mRNA expression of caspase 3 and caspase 9, and the protein expression of p53 upregulated modulator of apoptosis (PUMA) and Bcl-2-associated X protein (Bax), which are involved in apoptosis, increased with increasing OTA concentration; however, the protein levels decreased only at a high concentration (500 nM) (Figure 1b,c). Because the increased expression of the above factors does not mean that apoptosis occurs, we performed the TUNEL assay at three concentration conditions (0, 100, and 500 nM) to confirm DNA damage and apoptosis (Figure 1d).

### 2.2. OTA Triggers Oxidative Stress

Hepatocellular toxicity, including DNA damage and apoptosis induced by OTA, is attributed to oxidative stress [25]. To determine whether oxidative stress is induced by the concentration of OTA used, we measured the intracellular generation of ROS by using the fluorescent dye, dichlorofluorescein diacetate (DCFH-DA). We showed that ROS production increased in a dose-dependent manner following treatment with OTA. ROS production increased significantly in cells treated with OTA at 100 nM (Figure 2a). MDA, which is another indicator of oxidative stress, is a lipid peroxidation product and a carbonyl compound produced by externally-induced oxidative stress [28]. NO is also released in response to oxidative stress and can be used as an index of oxidative stress [29,30]. The thiobarbituric acid-reactive substances (TBAR) assay revealed that TBAR-reactive substances increased in a dose-dependent manner and NO production also increased in a dose-dependent manner (Figure 2b,c). The concentration of GSH, which an endogenous antioxidant, significantly decreased following treatment with OTA at 500 nM (Figure 2d). Increased expression of the biomarkers TBAR-reactive substances and NO indicates the occurrence of oxidative stress due to ROS generation and GSH reduction.

### 2.3. OTA Activates the Nrf2-ARE Signaling Pathway

To determine whether the Nrf2-ARE signaling pathway, which is a defense mechanism against oxidative stress [23], is activated by OTA, mRNA and protein levels of Nrf2 were measured. As the concentration of OTA increased, the mRNA and protein expression of Nrf2 increased in a dose-dependent manner (Figure 3). In particular, nuclear Nrf2 increased while there was no significant difference in cytosolic Nrf2, indicating that OTA induced the translocation of Nrf2 from the cytoplasm to the nucleus (Figure 3b,c). We also confirmed the expression of antioxidant enzymes involved in the phase II reaction. The expression of mRNA and proteins of the downstream antioxidant enzymes, HO-1 and GCLC, increased in a dose-dependent manner (Figure 4a,b), although GCL protein level decreased at high concentration.

### 2.4. OTA Is Metabolized through the CYP Family

Most drugs and toxic compounds undergo oxidation in the phase I reaction [31]. Similar to the results of previous studies [17], we also confirmed that OTA was metabolized via the phase I reaction. Among the CYP family isoforms involved in the reaction, CYP1A1/1A2 and CYP3A4 are abundant in hepatocytes, and CYP1A1/1A2 is regulated by AhR, whereas CYP3A4 is regulated by PXR [16]. The level of mRNA expression of AhR and PXR was significantly increased above 250 nM. They migrated from the cytoplasm to the nucleus in the case of AhR and PXR, which are transcription factors when activated, thus confirming the amount of protein expression in the nucleus extract. As a result, the protein expression of AhR and PXR in the nucleus was also increased compared to the control. (Figure 3). Therefore, the mRNA expression level of the CYP family affected by each transcription factor also significantly increased above 250 nM (Figure 4b). However, in the case of protein expression, CYP1A1 and CYP3A4 decreased at 250 nM and CYP1A2 decreased at 500 nM (Figure 4d). This seems to be due to inhibition of protein synthesis and expression [32] at high concentrations of OTA as shown in other studies. 

### 2.5. OTA-Induced DNA Damage through the Phase I Reaction and Protected DNA Damage through Nrf2-Signaling Pathway in the Phase II Reaction

The major transcription factors, AhR and Nrf2, which activate the phase I and II reactions, respectively, were silenced in HepG2 cells following siRNA transfection. As shown in Figure 5a, siAhR and siNrf2 effectively silenced AhR and Nrf2 at the mRNA level with efficiencies of 75 and 45%, respectively. Figure 5b shows that AhR silencing decreased the expression of CYP1A1/1A2 in cells treated with OTA. AhR-silencing also decreased the expression of Nrf2, which is a transcription factor for phase II enzymes, including HO-1 and GCLC, in cells treated with OTA (Figure 5c). In other words, when the expression of AhR was suppressed, the expression of Nrf2 and its target genes, HO-1 and GCLC, decreased together, so that AhR, the regulator of the phase I reaction, also regulates the phase II reaction.

We further investigated the effect of OTA-induced oxidative stress and Nrf2 on hepatotoxicity. As shown in Figure 6a,c, the mRNA expression of caspase 3 and caspase 9, and the amount of ROS decreased after AhR silencing. These results indicated that ROS is generated through the phase I reaction by CYP enzymes, resulting in DNA damage and apoptosis. Conversely, Figure 6b shows that Nrf2 silencing increased the degree of apoptosis by increasing the mRNA expression of caspase 3 and caspase 9. This suggested that DNA damage and apoptosis occurred because oxidative stress was not decreased due to the inactivation of the phase II enzymes by Nrf2, which is a major component of the antioxidant mechanism. Based on these results, OTA is predicted to induce oxidative stress in the phase I reaction, which leads to the activation of phase II enzymes through the antioxidant Nrf2-signaling pathway, thereby alleviating OTA-induced cytotoxicity.

### 2.6. In Vivo Phase I and Phase II Reactions Following OTA Administration

Animal experiments were conducted to confirm the trends observed in the in vitro experiments. Real-time polymerase chain reaction (PCR) and Western blotting were performed using liver tissues of imprinting control region (ICR) mice following oral administration of low (0.2 mg/kg BW), medium (1 mg/kg BW), and high (3 mg/kg BW) concentrations of OTA for six weeks. The results showed that the mRNA and protein expression of both phase I and phase II enzymes increased in a dose-dependent manner (Figure 7), as observed in the in vitro study. In particular, Western blot analysis revealed that the increase in protein expression at low concentrations, and the decrease in protein expression at higher concentrations were also consistent with the results of in vitro studies. 

## 3. Discussion

In vivo models and in vitro hepatic cell lines and primary hepatocytes demonstrate different responses under toxic conditions [33]. The activities of some metabolic enzymes in HepG2 cells are lower than those in primary hepatocytes; therefore, HepG2 cells may not be optimal for toxicity studies related to metabolism [34,35]. However, in the present study, we confirmed the toxicity of OTA in both in vitro and in vivo experiments to compensate for the limitations of HepG2 cells and to improve the reliability of our results. Previous studies have demonstrated the oxidative stress-induced toxicity of OTA in hepatocytes at a concentration of 5–800 µM [36] resulting in ROS and DNA damage. However, considering that the regulation standard for OTA in food is 1–50 ppb [37], our study was conducted using relatively lower nano-molar levels of OTA. Our experiments showed oxidative stress-induced cytotoxicity and apoptosis in HepG2 cells treated with OTA (5–500 nM). OTA has been reported to induce necrosis as well as apoptosis [27]. In particular, treatment with OTA at a low concentration activates the apoptotic process more than the necrotic process, whereas treatment with OTA at high concentrations, induces necrotic cell death. In addition, the induction of cell death via necrosis and apoptosis is cell-type-dependent [38]. In the present study, the green fluorescence, due to Alexa Fluor^TM^ 488 labelling, increased with increasing OTA concentrations (Figure 1d), indicating that apoptosis occurred due to DNA damage with OTA treatment at nano-molar levels in HepG2 cells.

Intracellular generation of the oxidative stress biomarkers, ROS, lipid peroxides, and NO increased with OTA treatment. Conversely, in our study, GSH, which is an antioxidant that protects cells and tissues from oxidative stress [39], tended to decrease with higher concentrations of OTA. This is because GSH is expected to be oxidized to GSSG, acting as a defense mechanism against oxidative stress. Another possible explanation for the decrease in GSH is that OTA-quinone, a metabolite of OTA, forms a conjugate with GSH during detoxification [39,40], and thus reduces the amount of GSH. ROS is produced in mitochondria and endoplasmic reticulum [41], but the mechanism of ROS formation induced by OTA has not yet been established. Other studies have shown that ROS is produced during the phase I reaction [42,43], which is an oxidation reaction, during the metabolism of toxic substances. Therefore, ROS is produced when OTA reacts with the phase I enzymes in the liver. We confirmed that ROS production decreased following the silencing of AhR, a transcription factor for phase I enzymes. Although the underlying mechanism of ROS production and GSH depletion remains unclear, these results tend to shift the balance of an intracellular redox balance toward oxidative profile.

Consistent with previous studies [44], we demonstrated the biotransformation of OTA during phase I and phase II enzymatic reactions. Particularly, we showed that mRNA and protein expression of CYP1A1/1A2 and 3A4, which are abundant in liver tissues [45,46], gradually increased in a dose-dependent manner. Of note, when treated with high concentrations of OTA (500 nM), mRNA expression increased while protein expression decreased. This may be because OTA inhibits synthesis of the protein, as previously shown in other studies [32,47]. As previously reported, phenylalanine is a component of the structure of OTA [48], and phe-tRNA synthetase uses OTA as a substrate instead of phenylalanine to inhibit the synthesis of tRNA [49], thereby reducing protein synthesis. The isoforms used in this experiment, including CYP1A1/1A2 and 3A4, are involved in the oxidation of OTA to produce 4 (R) - and 4 (S) -OH-OTA [17]. In addition, they help to reduce toxicity by forming metabolites, such as OTA-quinone, to form conjugates with GSH [39,43]. Conversely, CYP1A2 and 3A4 increase cytotoxicity [42]. Therefore, the dose-dependent increase in CYP1A1/1A2 and 3A4 expression observed in the present study suggest that OTA is metabolized through the phase I reaction in the liver, resulting in the production of various metabolites and ROS.

In response to this oxidative stress, defense mechanisms are activated, promoting adaptation and survival in response to oxidative stress. Nrf2-ARE is the most widely-studied signaling pathway, which regulates the expression of antioxidant enzymes in response to oxidative stress [50]. Our results showed that the mRNA and protein levels of Nrf2, HO-1, and GCLC increased with increasing concentrations of OTA; however, at 500 nM, protein expression decreased. Therefore, we confirmed that the phase II enzymes are activated as a defense mechanism against OTA-induced oxidative stress. 

Interestingly, previous studies have shown that OTA acts as an Nrf2 inhibitor through mechanisms such as miR132 induction, nrf2 nuclear translocation block, and Nrf2 transcription inhibition, thereby reducing the expression of Nrf2 [51]. However, in this study, the expression of Nrf2 and phase II enzymes was increased. This is probably because the phase II reaction, which provides antioxidant protection, was further activated by ROS and active intermediates generated through the phase I reaction in this study. In particular, since the OTA concentration in this study is lower than other studies, it seems that the antioxidant reaction is preferentially caused by the detoxifying effect on toxicity rather than the phenomenon caused by strong toxicity. 

We transfected cells in order to silence genes encoding transcription factors of phase I/II genes to confirm that the phase I reaction precedes the phase II reaction, and to determine how both reactions affect the toxicity of OTA in HepG2 cells. Our results indicated that the OTA-induced DNA damage and apoptosis occur due to the phase I reaction via CYP enzymes activated by AhR. In addition, AhR silencing resulted in decreased expression of Nrf2 and phase II enzymes, indicating that the phase II reaction is not activated in the absence of the phase I reaction due to the decreased expression of AhR. Thus, when the OTA is metabolized in the liver, the phase II reaction follows the phase I reaction, and the former is activated by metabolites generated during the latter reaction. The increased expression of caspase 3 and caspase 9 as a result of Nrf2 silencing indicates that Nrf2 reduces the hepatotoxicity of OTA by activating the phase II reaction.

## 4. Materials and Methods

### 4.1. Materials

OTA was supplied by Cfm Oskar Tropitzsch (Marktredwitz, Germany). Dulbecco′s modified Eagle′s medium (DMEM) and Opti-MEM were from Life Technologies (Grand Island, NY, USA). Penicillin-streptomycin, fetal bovine serum (FBS), and trypsin were purchased from Hyclone (Logan, UT, USA). The bicinchoninic acid (BCA) protein assay kit was purchased from Thermo Scientific (Rockford, IL, USA). In addition, MTT, dimethyl sulfoxide (DMSO), GSH, 5,5′-dithiobis (2-nitrobenzoic acid) (DTNB), glutathione reductase from baker′s yeast (GR), β-nicotinamide adenine dinucleotide 2′-phosphate reduced tetrasodium salt hydrate (β-NADPH), 2′,7′-dichlorodihydrofluorescein diacetate (DCFA-DH), 2,3-diaminonaphthalene (DAN), and nitric oxide (NO) were purchased from Sigma-Aldrich (St. Louis, MO, USA). The enhanced chemiluminescence (ECL) detection kit was purchased from Abclon (Seoul, South Korea). All of the antibodies used in the experiments were purchased from Santa Cruz Biotechnology (Santa Cruz, CA, USA). The glyceraldehyde-3-phosphate dehydrogenase (GAPDH) antibody was provided by Millipore (Termecular, CA, USA).

### 4.2. Cell Culture

Human hepatocyte HepG2 was purchased from the American Type Culture Collection (ATCC) (Manassas, VA, USA) and cultured in low-glucose DMEM supplemented with 10% FBS and 1% penicillin-streptomycin (37 °C, 5% CO_2_). All cells in the experimental procedure were sub-cultured when grown to a density of 80–90%, and only cells under passage 30 were used.

### 4.3. Measurement of Reactive Oxygen Species (ROS)

To determine the amount of ROS produced from HepG2 cells, cells were seeded in a 96-well plate. After 24 h, the supernatant was removed, and the cells were incubated for 30 min with 100 μM of 2′,7′-dichloro-dihydro-fluorescein diacetate (DCFH-DA). The supernatant was removed, washed with PBS and treated with OTA for 48 h. Fluorescence was measured using a multi-detection microplate reader (HIDEX) (Ex 485 nm/Em 535).

### 4.4. Measurement of Nitric Oxide (NO)

The amount of NO produced by HepG2 cells was measured via a fluorometric NO assay, with NO^2−^ present in the cell culture medium. HepG2 cells were seeded in a 24-well plate and treated with OTA. After 48 h, the cell culture supernatant was collected. Then, 10 μL of DAN working solution (2,3-diaminonaphthalene 50 μg/mL in 0.62 N HCl) and 5 μL of 2.8 N NaOH were added to 100 μL of the supernatant and fluorescence was measured (Ex 365 nm/Em 410 nm). The NO concentration in the cell culture medium was calculated based on the quantitative curve of sodium nitrite standard solution.

### 4.5. Glutathione Assay

To measure glutathione (GSH) contents, HepG2 cells were seeded in a 6-well plate and cultured for 24 h, and then treated with OTA for 48 h. The cells were collected in 100 μL of 5% sulforsalisylic acid and then disrupted using an ultrasonicator. In the experiment, the supernatant was obtained by centrifuging the disrupted cells at 12,000× *g*, 30 min, and 4 °C. Then, 140 μL of cocktail solution (20 μL of 0.1 M KPE, 60 μL of 1.68 mM DTNB, and 60 μL of 3.3 U/mL GR) was added to 20 μL of the supernatant and reacted for 30 s. Then, 60 μL of 0.89 mM β-NADPH was added and the absorbance was measured.

### 4.6. TBAR Assay 

The amount of TBAR-reactive substances was measured using the TBARs assay. HepG2 cells were seeded in a 6-well plate and cultured for 24 h, followed by treatment with OTA for 48 h. Cells were collected in 200 μL of homogenization buffer (butylated hydroxytoluene (BHT) 8 mg/200 mL of PBS), homogenized using an ultrasonicator, and centrifuged at 12,000× *g*, 30 min, 4 °C to obtain supernatant. Then, 1 mL of TBA reagent was added to 100 μL of supernatant, boiled at 100 °C for 15 min, and then centrifuged at 10,000 rpm for 5 min. Supernatant (50 µL) was dispensed into a 96-well plate and the fluorescence (Ex 530/Em 590) was measured. 

### 4.7. Total RNA Isolation and cDNA Synthesis

HepG2 cells were seeded in a 6-well plate. After incubation for 24 h, cells were treated with OTA for 48 h, after which RNA was isolated with 500 μL of RNAiso Plus (TaKaRa Co., Ltd., Kusatsu, Japan) per well. The amount of RNA isolated was confirmed by NanoDrop^TM^2000 (Thermo Scientific, Rockford, IL, USA). cDNA was synthesized from RNA using a first-strand cDNA synthesis kit (LeGene Biosciences, San Diego, CA, USA).

### 4.8. Real-Time PCR

Quantitative real-time PCR (qPCR) was performed using 1 μL of cDNA with 9 μL of primer and 10 μL of SYBR green in a total reaction volume of 20 μL on an IQ5 real-time PCR system (Bio-Rad). The types and sequences of human primers used in an in vitro study were shown in Appendix A. GAPDH was used as a house keeping gene. The results were analyzed using the comparative C_T_ method as a means of relative quantification, normalized to the housekeeping gene (GAPDH) and expressed as 2^−△△CT^ values. Melting curve analysis was performed to assess product specificity.

### 4.9. Isolation of Nuclear and Cytosolic Extracts

After OTA treatment, HepG2 cells were washed with PBS, collected in 500 μL of PBS, and centrifuged at 13,000× *g* for 3 min to remove the supernatant. The pellet was resuspended in hypotonic buffer A (10 mM HEPES, 10 mM KCl, 0.1 mM EDTA, 1 mM DTT, 1 mM PMSF, 5 μg/mL Leupeptin and Aprotinin; pH 7.8), and 0.8% NP-40 was added, followed by centrifugation at 12,000× *g* for 5 min at 4 °C. The supernatant was used as a cytosolic solution. Pellet was resuspended with hypotonic buffer A, and 0.8% NP-40 was added. The supernatant was removed by centrifugation at 12,000× *g*, 5 min, and 4 °C. Pellet was dissolved in Buffer B (50 mM HEPES, 50 mM KCl, 300 mM NaCl, 0.1 mM EDTA, 20% Glycerol, 1 mM DTT, 1 mM PMSF, 5 μg/mL Leupeptin, Aprotinin; pH 7.8) and centrifuged at 12,000× *g* for 5 min at 4 °C. The supernatant was used as nuclear.

### 4.10. Isolation of Total Cell Lysates and Western Blot Analysis

After treatment with OTA, HepG2 cells were washed with PBS, and then collected by the addition of radioimmunoprecipitation assay (RIPA) buffer. The supernatant, obtained by shaking for 15 min and centrifuging at 13,000 rpm for 20 min, was used. Sodium dodecyl sulfate polyacrylamide gel electrophoresis (SDS-PAGE) was performed with the same amount of protein (64 μg) and transferred to polyvinylidene (PVDF) membranes (Millipore, Billerica, MA, USA). Membranes were blocked with blocking solution (5% skimmed milk in tris buffered saline with Tween-20 (TBST) for 1 h and then incubated at 4 °C with primary antibody. After the primary antibody was recovered, the membranes were washed and incubated with the secondary antibody at room temperature for 45 min. The results were observed with ECL solution and the band was quantified by ImageJ software (version, National Institutes of Health, Bethesda, MD, USA). The values of the bands obtained by using the ImageJ software were divided by the values of housekeeping gene and corrected. The control group was set to 1 and the other groups were compared with the control group.

### 4.11. Transfection of siRNA

Reverse transfection was performed using Lipofectamine^TM^ RNAiMAX (Invitrogen, Carlsbad, CA, USA). siRNA (30 pmol; Shanghai GenePharma Co., Ltd., China) and 100 μL of Opti-MEM were dispensed into 24-well plates, and 1 μL of lipofectamine^TM^ RNAiMAX was added to each well. After incubation for 15 min, HepG2 cells were seeded at a density of 1.5 × 10^5^ cells/mL. The siRNA sequences for AhR and Nrf2 were as follows: AhR, 5′-GCCUGUAUUACCACAACAUTTAUGUUGUGGUAAUACAGGCTT-3′;Nrf2, 5′-GGCAUAGAGACCGACUUAATTUUAAGUCGGUCUCUAUGCCTT-3′

### 4.12. TUNEL Assay

To confirm that apoptosis and DNA damage had occurred, the TUNEL assay was performed using the APO-BrdU^TM^ TUNEL Assay Kit (Invitrogen, Carlsbad, CA, USA). HepG2 cells were seeded in a 60 mm dish at a density of 1 × 10^6^ cells/mL. The subsequent steps were performed according to the protocol recommended in the kit. Data analysis was performed using a confocal laser scanning microscope (CLSM) (Carl-Zeiss, Oberkochen, Germany).

### 4.13. Animal Study

#### 4.13.1. Test Animals and Breeding Environment

In this study, 7-week-old male ICR-type mice purchased from Orient Bio Co., Ltd. were used, and only healthy mice were selected for testing after 1 week of quarantine and purification. The experimental animals were maintained in an environment set at 20 ± 3 °C, a relative humidity of 50 ± 10%, ventilation frequency of 10–20 times/h, and a lighting time of 12 h (8:00 a.m.–8:00 p.m.). During the entire study period, water and AIN-93 standard diets were provided ad libitum, and dietary intakes and body weights (BW) were measured once each week. This study was carried out in accordance with the guidelines of the Committee for Ethical Usage of Experimental Animals of Korea University (KUIACUC-2018-1, 8 January 2018).

#### 4.13.2. Sample Administration

All animals were weighed and divided into five groups of eight animals by random sampling. The control groups included are normal control and a solvent control group. The treatment group received ochratoxin 0.2 mg/kg BW (Group 1), 1 mg/kg BW (Group 2), and 3 mg/kg BW (Group 3). The samples were orally administered for 5 days every week for 6 weeks from the start of the experiment. The dose (100 μL/10 g) was calculated based on the most recently measured BW. After sacrifice, a portion of the mouse liver was stored in TRIzol and the remaining portion was stored in PBS.

### 4.14. Statistical Analysis

All results are expressed as the mean ± standard deviation (SD) Three repeated experiments were performed for each experiment (*n* = 3). Different letters indicate significant differences at *p* < 0.05 by ANOVA with Tukey’s multiple range tests. If the same letters exist for each group, there is no statistically significant difference, and if there are no letters, it means that there is a statistically significant difference. All statistical analyses were performed using SAS version 9.4 (SAS institute, Cary, NC, USA).

## 5. Conclusions

Studies using HepG2 cells and ICR mice demonstrated that OTA induces oxidative stress through the phase I reaction, and that the phase II reaction, activated by the Nrf2-ARE signaling pathway in response to oxidative stress, plays an important role in reducing OTA-induced hepatotoxicity including DNA damage and apoptosis. Therefore, our study elucidates the role of the phase I and II reactions through transcription factor AhR and Nrf2 in OTA-induced oxidative stress and confirms the relationship between both reactions at very low concentrations. In particular, it is the significance of this study that the most important factor in the hepatotoxic mechanism of OTA is AhR, the transcription factor regulating phase I and phase II reactions. This indicates the possibility that regulation of AhR may be applied in studies on the prevention and treatment of OTA-induced hepatotoxicity.

## Figures and Tables

**Figure 1 toxins-11-00377-f001:**
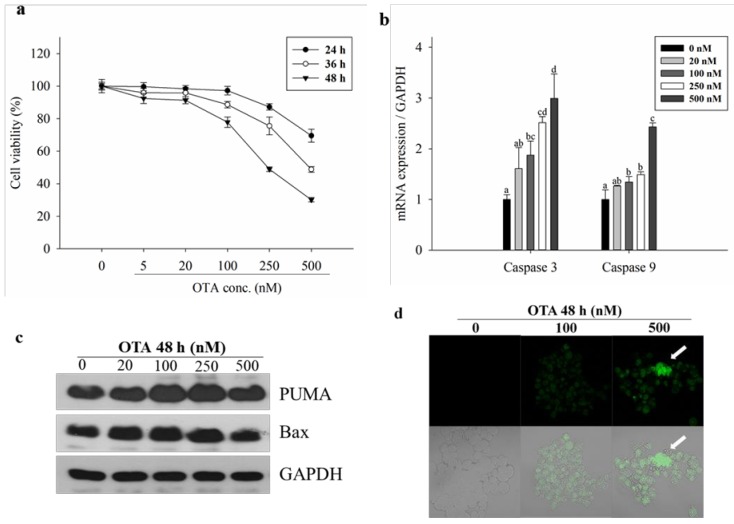
Effect of ochratoxin A (OTA) on cytotoxicity and apoptosis in HepG2 cells. (**a**) cells were treated with various concentrations of OTA (5, 20, 100, 250, 500 nM) for 24, 36, 48 h, followed by determination of cell viability by 3-(4,5-Dimethyl-2-thiazolyl)-2,5-diphenyl-2H-tetrazolium bromide (MTT) assay; (**b**) mRNA expressions of caspase3, caspase9 were measured by quantitative real-time polymerase chain reaction (PCR); (**c**) total cellular protein levels of p53 upregulated modulator of apoptosis (PUMA), Bcl-2-associated X protein (Bax) were detected by Western blotting. Glyceraldehyde-3-phosphate dehydrogenase (GAPDH) was used as a housekeeping gene; (**d**) HepG2 cells treated with OTA for 48 h and stained using the APO-BrdU^TM^ terminal deoxynucleotidyl transferase dUTP nick end labeling (TUNEL) Assay Kit (A23210). Cells containing DNA strand nicks characteristic of apoptosis are detected by TUNEL and green fluorescence. Representative photomicrographs for untreated control cells, cells treated with 100 nM OTA and 500 nM OTA are shown (magnification 400×). The upper panel shows only green fluorescence and the lower panel shows cell morphology with green fluorescence. Data are expressed as mean ± standard deviation (SD) of three independent experiments. Different letters indicate significant differences at *p* < 0.05 by Tukey’s multiple range test.

**Figure 2 toxins-11-00377-f002:**
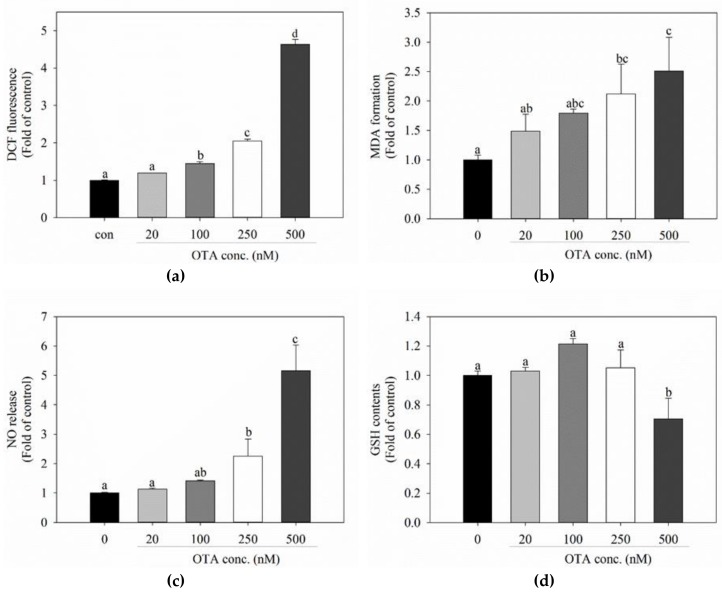
OTA triggers oxidative stress in HepG2 cells. (**a**) levels of reactive oxygen species were measured 48 h after exposure to various concentrations (0, 20, 100, 250, and 500 nM) of OTA by H2DCFDA staining; (**b**) malondialdehyde levels were measured 48 h after exposure to various concentrations of OTA; (**c**) nitric oxide release was measured 48 h after exposure to various concentrations of OTA by Griess reagent; (**d**) total intracellular glutathione levels were measured after 48 h of OTA treatment. Data are expressed as the mean ± SD of three independent experiments. Different letters indicate significant differences at *p* < 0.05 by Tukey’s multiple range test.

**Figure 3 toxins-11-00377-f003:**
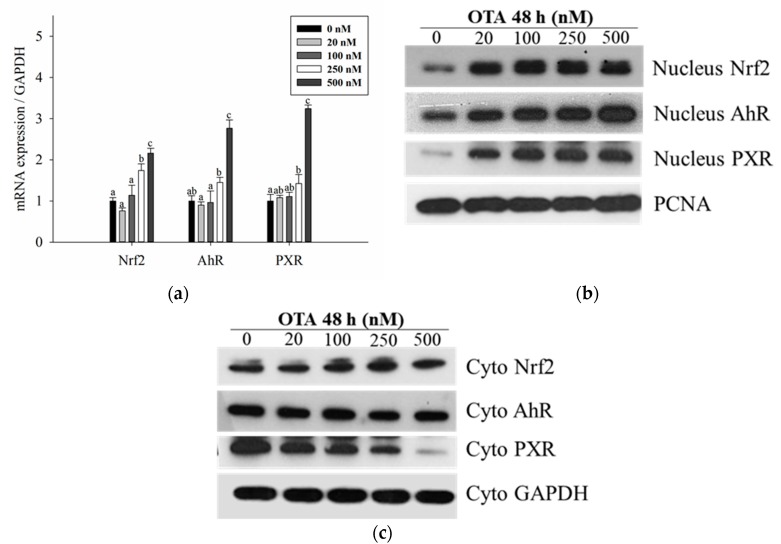
Activation of transcription factors of phase I and phase II enzymes by OTA. All experiments were carried out under the same conditions and with treatment at various concentration (0, 20, 100, 250, and 500 nM) for 48 h. (**a**) mRNA expression of transcription factors (Nrf2, AhR, and PXR) was measured by quantitative real-time PCR; (**b**) detection of Nrf2, AhR, and PXR in nuclear protein fractions by Western blotting. PCNA was used as a housekeeping gene; (**c**) detection of Nrf2, AhR, and PXR in cytoplasmic protein fractions by Western blotting. GAPDH was used as a housekeeping gene. Data are expressed as the mean ± SD of three independent experiments. Different letters indicate significant differences at *p* < 0.05 by Tukey’s multiple range test.

**Figure 4 toxins-11-00377-f004:**
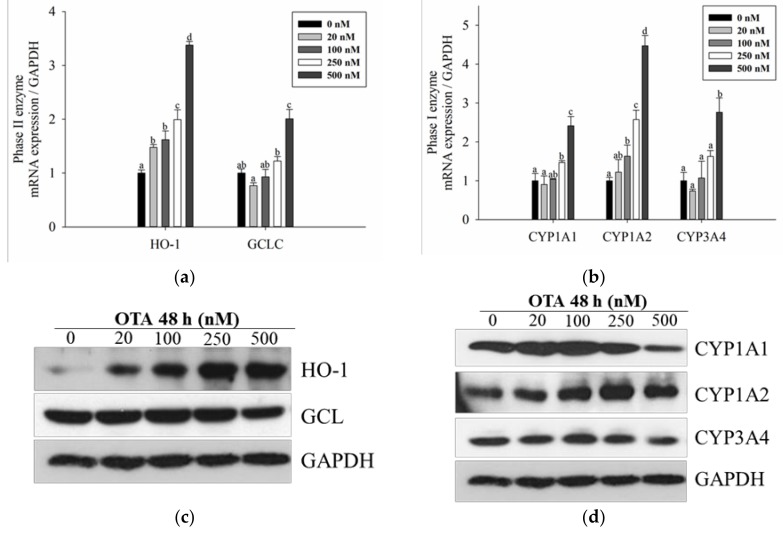
OTA activates phase I and II pathway. (**a**) mRNA expression and (**b**) total protein level of phase II enzymes (HO-1: heme oxygenase-1 and GCL/GCLC: glutamate-cysteine ligase/GCL catalytic subunit) were measured at 48 h after OTA treatment; (**c**) mRNA expression and (**d**) total protein level of phase I enzymes (CYP1A1, CYP1A2, and CYP3A4) were measured at 48 h after OTA. mRNA expression and total protein level were measured by quantitative real-time PCR and Western blotting. GAPDH was used as a housekeeping gene. Data are expressed as mean ± SD of three independent experiments. Different letters indicate significant differences at *p* < 0.05 by Tukey’s multiple range test.

**Figure 5 toxins-11-00377-f005:**
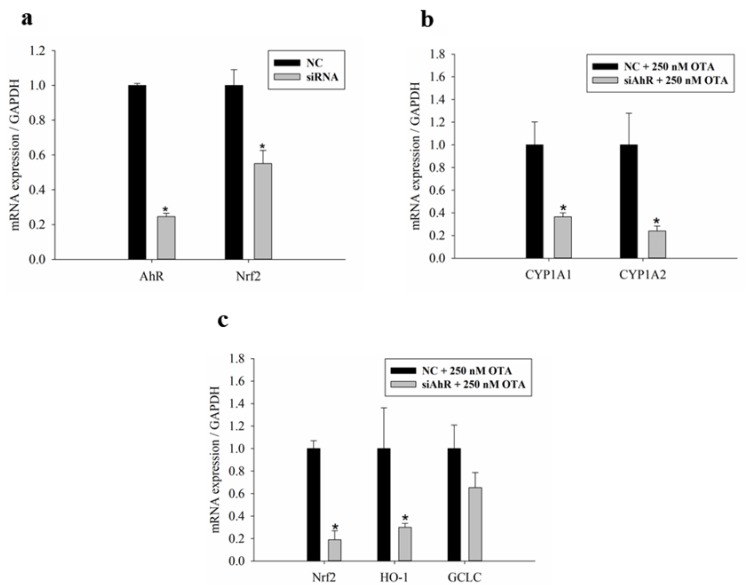
Silencing of transcription factors regulates phase I and II pathway. HepG2 cells are transfected with siAhR or siNrf2 for 24 h, and then cultured for 48 h with OTA. (**a**) the efficiency of AhR and Nrf2 knockdown was determined by real-time PCR. After silencing of AhR, the expression levels of (**b**) phase I and (**c**) phase II enzymes were detected by real-time PCR. Data are expressed as the mean ± SD of three independent experiments. * *p* < 0.05 siRNA compared with the negative control (NC).

**Figure 6 toxins-11-00377-f006:**
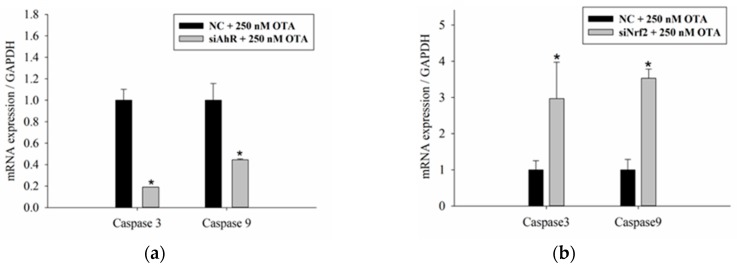
Effects of AhR and Nrf2 silencing on oxidative stress and apoptosis. HepG2 cells were transfected with (**a**) siAhR or (**b**) siNrf2 for 24 h, and then cultured for 48 h with 250 nM OTA. The expression levels of caspase 3 and caspase 9 were determined by real-time PCR. (**c**) Following AhR silencing and 500 nM OTA treatment, ROS levels were measured by H_2_DCFDA staining. Data are expressed as the mean ± SD of three independent experiments. * *p* < 0.05 siRNA compared with the NCs.

**Figure 7 toxins-11-00377-f007:**
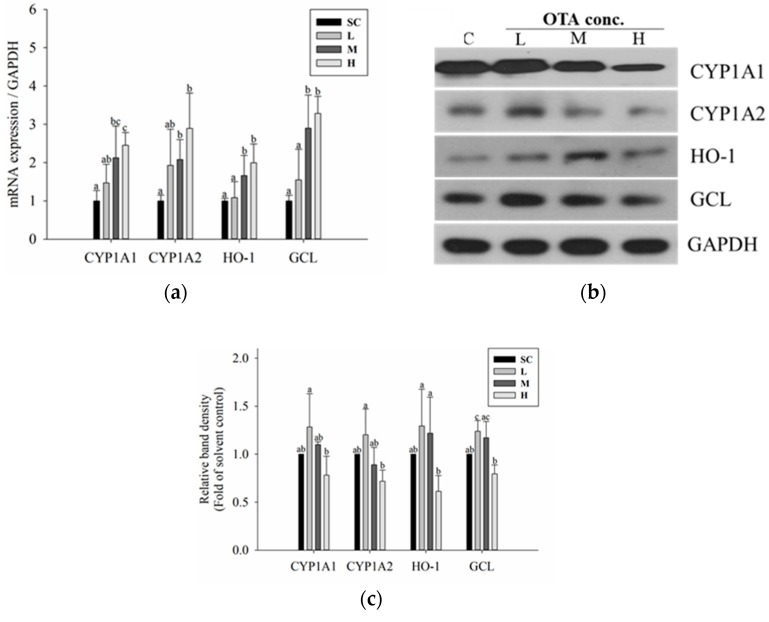
OTA activates phase I and phase II pathways in vivo. (**a**) mRNA expressions of phase I (CYP1A1 and CYP1A2) and phase II (HO-1 and GCL) enzymes were measured by quantitative real-time PCR; (**b**) total cellular protein levels of phase II enzymes (HO-1 and GCL) were detected by Western blotting. GAPDH was used as a housekeeping gene; (**c**) intensities of bands of phase I and II enzymes were normalized to that of GAPDH. Data are expressed as the mean ± SD of three independent experiments. Different letters indicate significant differences at *p* < 0.05 by Tukey’s multiple range test.

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
