# Peer review of "Ochratoxin A-Induced Hepatotoxicity through Phase I and Phase II Reactions Regulated by AhR in Liver Cells"

_toxins, 2019, doi:10.3390/toxins11070377_

Round 1
Reviewer 1 Report
In general, the paper is well written and gives new findings on the mechanism of cellular toxicity of ochratoxin A. There are some minor points to be corrected.
It is not indicated everywhere how many independent experiments and technical replicates were done that is essential for statistical analysis.
It is not clear how the western blot data were interpreted: simple densitometry or ECL ratios (signals referred to control bands).
In Fig. 6. the pdf visualized the data incorrectly.
It is not clear that if the cells were scraped then the data were referred to total protein or to some other reference.
The TBARs assay is not generally accepted to give precise information on the MDA levels. It is better to say "TBAR-reactive substances". For correct quantification an MDA Elisa assay is required.
Author Response
Reviewer(s’) comments
Comments
1) It is not indicated everywhere how many independent experiments and technical replicates were done that is essential for statistical analysis.
Answer : We appreciate the reviewer’s comment. The concerned corrections were highlighted with yellow color in text.
In the legendary, we have already written 'Data are expressed as mean ± SD of three independent experiments'. However, we added the following sentence to Material & Methods for easier identification.
Three repeated experiments were performed for each experiment. (Line 518-519)
2) It is not clear how the western blot data were interpreted: simple densitometry or ECL ratios (signals referred to control bands).
Answer : We appreciate the reviewer’s comment.
In order to more specifically describe the interpretation of the Western blot results, we modified the contents of that part of Material & Methods into the following sentence.
The results were observed with ECL solution and the band was quantified by Image J software (National Institutes of Health, Maryland, U.S.A.). The values of the bands obtained by using the Image J software were divided by the values of housekeeping gene and corrected. The control group was set to 1 and the other groups were compared with the control group. (Line 475-479)
3) In Fig. 6. the pdf visualized the data incorrectly.
Answer : We appreciate the reviewer’s comment. We rearranged the figure.
4) It is not clear that if the cells were scraped then the data were referred to total protein or to some other reference.
Answer : We appreciate the reviewer’s comment.
We modified that part to avoid confusion.
The cells were collected in 100 μL of 5% sulforsalisylic acid and then disrupted using an ultrasonicator. (Line 425-426)
Cells were collected in 200 μL of homogenization buffer (BHT 8 mg/200 mL of PBS), homogenized using an ultrasonicator, and centrifuged at 12,000 × g, 30 min, 4°C to obtain supernatant. (Line 434-436)
5) The TBARs assay is not generally accepted to give precise information on the MDA levels. It is better to say "TBAR-reactive substances". For correct quantification an MDA Elisa assay is required.
Answer : We appreciate the reviewer’s comment.
In the text, we have modified 'MDA' to ‘TBA-reactive substances’ in Line 127, 130, 433 as follows
The TBAR assay revealed that TBAR-reactive substances increased in a dose-dependent manner and NO production also increased in a dose-dependent manner (Figs. 2b and c).
Increased expression of the biomarkers TBAR-reactive substances and NO indicates the occurrence of oxidative stress due to ROS generation and GSH reduction.
The amount of TBAR-reactive substances was measured using the TBARs assay.

Reviewer 2 Report
The author used different letters show the significant differences, it caused confusion. Please explain it clearly.
The conclusions need to preside. The author claimed that “AhR and PXR mRNA and protein expression increased with OTA treatment”(Line 178). However, at the transcript level, AhR and PXR expression didn’t increase with 20 or 100 nM OTA treatment. At the protein level, there is no data to support that AhR or PXR expression increased.
The author claimed that “protein expression of CYP1A1/1A2 and CYP3A4 increased in a dose-dependent manner” (Line 179). However, CYP1A1 and CYP3A4 expression didn’t support the conclusion.
Fig.5 Fig.6 and Fig.7 were not shown in the manuscript due to the composing.
Author Response
Reviewer 2:
Comments and Suggestions for Authors
1) The author used different letters show the significant differences, it caused confusion. Please explain it clearly.
Answer : We appreciate the reviewer’s comment. The concerned corrections were highlighted with yellow color in text.
In the Material & Methods part and each legendary, we wrote ‘Different letters indicate significant differences at p<0.05 by ANOVA with Tukey’s multiple range tests.’
In addition, we added a detailed description of the meaning of letters in the M & M part.
If the same letters exist for each group, there is no statistically significant difference, and if there is no letters, it means that there is a statistically significant difference.(line 520-522)
2) The conclusions need to preside. The author claimed that “AhR and PXR mRNA and protein expression increased with OTA treatment”(Line 178). However, at the transcript level, AhR and PXR expression didn’t increase with 20 or 100 nM OTA treatment. At the protein level, there is no data to support that AhR or PXR expression increased.
Answer : We appreciate the reviewer’s comment.
For a more specific statement, we fixed the contents as follows
The level of mRNA expression of AhR and PXR was significantly increased above 250 nM. They migrated from the cytoplasm to the nucleus in the case of AhR and PXR, which are transcription factors when activated, thus confirming the amount of protein expression in the nucleus extract. As a result, the protein expression of AhR and PXR in the nucleus was also increased compared to the control. (Fig. 3). (Line 181-185)
3) The author claimed that “protein expression of CYP1A1/1A2 and CYP3A4 increased in a dose-dependent manner” (Line 179). However, CYP1A1 and CYP3A4 expression didn’t support the conclusion.
Answer : We appreciate the reviewer’s comment.
For a more specific statement, we fixed the contents as follows
Therefore, the mRNA expression level of the CYP family affected by each transcription factor also significantly increased above 250 nM (Fig. 4b). However, in the case of protein expression, CYP1A1 and CYP3A4 decreased at 250 nM and CYP1A2 decreased at 500 nM (Fig. 4d). This seems to be due to inhibition of protein synthesis and expression [32] at high concentrations of OTA as shown in other studies.(Line 185-189)
4) Fig.5 Fig.6 and Fig.7 were not shown in the manuscript due to the composing.
Answer : We appreciate the reviewer’s comment. we rearranged the figures.
